# Trypsin/Zn_3_(PO_4_)_2_ Hybrid Nanoflowers: Controlled Synthesis and Excellent Performance as an Immobilized Enzyme

**DOI:** 10.3390/ijms231911853

**Published:** 2022-10-06

**Authors:** Zichao Wang, Pei Liu, Ziyi Fang, He Jiang

**Affiliations:** The Key Laboratory of Space Applied Physics and Chemistry, Ministry of Education, Shaanxi Key Laboratory of Macromolecular Science and Technology, School of Chemistry and Chemical Engineering, Northwestern Polytechnical University, Xi’an 710072, China

**Keywords:** hybrid materials, trypsin, nanoflowers, controlled synthesis, catalytic performance

## Abstract

Immobilized enzymes are a significant technological approach to retain enzyme activity and reduce enzyme catalytic cost. In this work, trypsin-incorporated Zn_3_(PO_4_)_2_ hybrid nanoflowers were prepared via mild precipitation and coordination reactions. The controllable preparation of hybrid nanoflowers was achieved by systematically investigating the effects of the raw-material ratio, material concentration and reaction temperature on product morphology and physicochemical properties. The enzyme content of hybrid nanoflowers was about 6.5%, and the maximum specific surface area reached 68.35 m^2^/g. The hybrid nanoflowers exhibit excellent catalytic activity and environmental tolerance compared to free trypsin, which was attributed to the orderly accumulation of nanosheets and proper anchoring formation. Further, the enzyme activity retention rate was still higher than 80% after 12 repeated uses. Therefore, trypsin/Zn_3_(PO_4_)_2_ hybrid nanoflowers—which combine functionalities of excellent heat resistance, storage stability and reusability—exhibit potential industrial application prospects.

## 1. Introduction

Protein–inorganic hybrid nanoflowers (PINFs), the integration of proteins and phosphate, a novel research field, have attracted great attention and have achieved rapid development since they emerged in 2012 [1,2,3,4,5]. Generally, PINFs can be classified according to the types of the inorganic component and the organic component [2,6,7]. A series of inorganic phosphate, such as zinc [7,8], copper [9,10], calcium [11,12], manganese [13,14] and cobalt [15,16] phosphate, have been employed to construct PINFs. Additionally, several organic biological macromolecules, including enzymes [17,18], proteins [19,20] and nucleic acids [21,22], have been reported for the synthesis of PINFs. PINFs are outstanding candidates for catalytic degradation, catalytic synthesis, detection, sensing, water treatment and other fields [1,23,24,25]. As a serine proteolytic enzyme, trypsin is the most specific protease and has an indispensable role in the amino acid arrangement of proteins [26]. Trypsin contains a large number of peptide bonds composed of basic amino acids, such as arginine and lysine, and is widely used in anti-inflammatory detumescence, leather manufacturing, raw silk processing and food processing. In general, free trypsin is stable in acidic conditions but sensitive to alkaline conditions. In order to solve the problem of separation and reuse of trypsin, several methods, including polymer composite microspheres [27], nano-composite membrane [28], mesopore silicon oxide [29], magnetic particles [30], natural polymer microspheres/fibers [31], quantum dots [32] and other immobilized carriers, have been selected to improve the stability of trypsin. Furthermore, the enzyme activity also improves after immobilization.

For example, Esmaeil Aslani and coworkers reported the preparation of immobilized trypsin by using Fe_3_O_4_@SiO_2_-NH_2_ [30]. The separation process of this material is simple and rapid and requires low energy consumption. Moreover, it has higher activity than free enzyme in different organic solvents. After six uses, the hydrolysis capacity of casein remained 85% of the initial activity. Chao Zhong and coworkers elegantly designed novel trypsin–copper phosphate hybrid nanoflowers containing copper phosphate as an inorganic compound and trypsin as the binding agent. As a micro-reactor for the hydrolysis of bovine serum albumin (BSA), the immobilized trypsin can be stored at 4 °C for 20 days and still maintain virtually undiminished catalytic activity [33].

Although it is possible to immobilize trypsin in the form of hybrid nanoflowers, the preparation generally takes a long time (1–3 days) when copper ions are used as coordination primitives. It has been confirmed in previous work that the creation of hybrid nanoflowers based on zinc phosphate is significantly faster than that of other inorganic components. The preparation time of BSA/Zn_3_(PO_4_)_2_, papain/Zn_3_(PO_4_)_2_ and lipase/Zn_3_(PO_4_)_2_ was about 3 h. Further, in contrast to free lipase, the catalytic activity of lipase/Zn_3_(PO_4_)_2_ was increased by 47%, which obviously surpassed lipase/Cu_3_(PO_4_)_2_. Thus, it is a great prospect to construct high-efficiency nano-flowers by selecting zinc phosphate. In this work, original hybrid trypsin/Zn_3_(PO_4_)_2_ nanoflowers were prepared quickly and efficiently. Meanwhile, we investigated the effects of reaction temperature and feed amount on the morphology of the nanoflowers and clarified the formation mechanism of the nanoflowers by tracking their morphology and composition changes. Using casein as the substrate, the catalytic activity of the trypsin/Zn_3_(PO_4_)_2_ hybrid nanoflowers was assessed, the catalytic conditions were optimized, and the stability of the nanoflowers for reuse was also studied.

## 2. Result and Discussion

### 2.1. Structural Characteristics of Hybrid Nanoflowers

The structure of prepared trypsin/Zn_3_(PO_4_)_2_ hybrid nanoflowers was characterized by Fourier transform infrared (FT-IR), X-ray powder diffraction (XRD), elemental analysis and thermogravimetric analysis (TGA), employed as shown in Appendix A. XRD was performed on trypsin, Zn_3_(PO_4_)_2_ and trypsin/Zn_3_(PO_4_)_2_ (Appendix A). The XRD spectrum of prepared Zn_3_(PO_4_) nanoparticles was in accordance with Zn_3_(PO_4_)_2_·4H_2_O (JCPDS, card no. 33-1474). Meanwhile, the XRD spectrum of the trypsin/Zn_3_(PO_4_)_2_ hybrid nanoflowers contained the diffraction peaks attributed to trypsin and Zn_3_(PO_4_)_2_ nanoparticles, certifying that Zn_3_(PO_4_)_2_ hybrid nanoflowers were synthesized successfully. Furthermore, the structures of trypsin, Zn_3_(PO_4_)_2_ and trypsin/Zn_3_(PO_4_)_2_ were characterized by FTIR (Appendix A). The typical characteristic peaks of trypsin stretching at 1400–1600 cm^−1^ for -NH_2_, 2800–3000 cm^−1^ for -CH_2_ and -CH_3_ and the peaks for Zn_3_(PO_4_)_2_ were observed in the FTIR spectrum of trypsin/Zn_3_(PO_4_)_2_. The results indicated the nanoflowers were constructed by trypsin and Zn_3_(PO_4_)_2_·4H_2_O. As shown in Appendix A, elemental analysis and thermal gravimetric analysis were used to characterize the trypsin/Zn_3_(PO_4_)_2_ to confirm the component contents of nanoflowers. The contents of C, H and N in hybrid nanoflowers were 2.565%, 2.238% and 0.647%, respectively (Appendix A). The N element only comes from trypsin, and the ratio of C and N elements in nanoflowers was consistent with the ratio of the two elements in trypsin. The higher H content was due to bound water in inorganic components. Furthermore, the results of TGA demonstrated that the organic composition in the nanoflowers was completely thermally decomposed at 350 °C, and the solid residue content was 93.42% under O_2_ atmosphere (Appendix A). Therefore, the content of trypsin in the trypsin/Zn_3_(PO_4_)_2_ hybrid nanoflowers was about 6.5%.

### 2.2. Controlled Synthesis of Hybrid Nanoflowers

The transformation of nanoflower morphology by adding trypsin amounts from 0 g to 0.25 g is shown in Figure 1. Without trypsin, Zn^2+^ reacted with PO_4_^−^ in PBS buffer solution to form Zn_3_(PO_4_)_2_ precipitation with a size of about 15 μm, demonstrating flower-like morphology assembled by lamellar (Figure 1a,b). With the increase of trypsin, the morphology of nanoflowers changed significantly: the particle size of the nanoflowers decreased (5–7 μm), the nanosheets were thinner, more nanosheets were needed to assemble nanoflowers and the interlayer spacing was reduced (Figure 1c–l). The mechanism for the above phenomenon is that trypsin acts as a crystal inhibitor during nanoflower formation, limiting nanosheet growth. Zn^2+^ interacted with the functional groups of the trypsin by electrostatic interaction and metal coordination, which promoted assembly between nanosheets. Further, as shown in Figure 1i–l, the morphological uniformity of nanoflowers was destroyed when the amount of trypsin reached 0.25 g; it can be seen that the particle size distribution becomes wider. This is due to fact that Zn^2+^ interacted with the functional groups of trypsin easily when trypsin was added in excess, which resulted in in situ formation of Zn_3_(PO_4_)_2_ on trypsin. Trypsin/Zn_3_(PO_4_)_2_ with larger freedom of movement assembled randomly. Thus, the particle size of trypsin/Zn_3_(PO_4_)_2_ hybrid nanoflowers was affected by the initial nucleation, and the secondary nucleation was more obvious.

Figure 2 demonstrates the N_2_ adsorption–desorption isotherms and pore size distributions of trypsin/Zn_3_(PO_4_)_2_ hybrid nanoflowers creating by adding 0 g to 0.25 g trypsin. The N_2_ adsorption–desorption isotherms were all assigned to type IV adsorption isotherms (isotherm with hysteresis loop) according to IUPAC with H3 hysteresis loop, indicating that the pore channels of the hybrid nanoflowers were narrow-slit pores with conical structure, which was consistent with the stacked morphology of lamellar structure observed by SEM. Further, it can be seen from the pore size distribution curve that mesoporous (2–50 nm) and macroporous (50–100 nm) structures were included in trypsin/Zn_3_(PO_4_)_2_ hybrid nanoflowers. Mesoporous structures were constructed from the tightly packed lamellar interior, and the macroporous structures were attributed to the dense accumulation of particles. Overall, increasing the amount of trypsin increased the number of mesopores and the specific surface area of the hybrid nanoflowers, reducing the average pore diameter (Appendix A).

The effects of Zn^2+^ dosage and system concentration on the morphology of trypsin/Zn_3_(PO_4_)_2_ hybrid nanoflowers were further investigated, as shown in Figure 3. It can be seen from the SEM images that hybrid nanoflowers with uniform morphology and narrow particle-size distribution were observed when the amount of Zn(OAc)_2_·2H_2_O was decreased from 0.56 g to 0.14 g (Figure 3a–h). With the reduction of Zn^2+^, the number of layers of the trypsin/Zn_3_(PO_4_)_2_ hybrid nanoflowers was increased, and the particle size did not change evidently. The morphologies of the nanoflowers prepared with different reactant concentrations were also observed. Multilayer trypsin/Zn_3_(PO_4_)_2_ hybrid nanoflowers prepared with high reactant concentration were formed at suitable raw material ratios (Figure 3i–l).

Crystal growth and assembly are directly affected by the reaction temperature. Thus, the morphology of trypsin/Zn_3_(PO_4_)_2_ hybrid nanoflowers prepared at different temperatures (30 °C, 40 °C and 50 °C) is shown in Appendix A. The particle size was not affected by the reaction temperature. The number of nanoplate layers of nanoflowers increased with temperature. However, the thickness of the nanoplates of nanoflowers decreased from 150 nm (30 °C) to 100 nm (50 °C). These results reveal that high temperature contributed to the rapid formation and assembly of nanosheets, so the number of nanoplate layers increased and the nanoplate layer thickness decreased.

### 2.3. Formation Mechanism of Trypsin/Zn_3_(PO_4_)_2_ Hybrid Nanoflowers

To explore in depth the morphological construction of trypsin/Zn_3_(PO_4_)_2_ hybrid nanoflowers and the formation mechanism, SEM images of nanoflowers in the reaction time range of 10 min to 120 min were observed serially, as shown in Figure 4. Nanoparticles of Zn_3_(PO_4_)_2_ were formed at 10 min (Figure 4a). When the reaction was carried out for 20 min, monolayer nanosheets were observed (Figure 4b). With the prolonged reaction, the monolayer nanosheets were assembled to the rudiment of hybrid nanoflowers with multilayer nanosheets at 60 min (Figure 4c). Continuing the reaction to 120 min, the thickness and particle size of the hybrid nanoflowers increased continuously, which exhibited that the assembly process was still ongoing (Figure 4d). Overall, these results suggest the mechanism for hybrid nanoflower self-assembly was crystal formation, nanosheet growth, nanosheet assembly and epitaxial growth. These process is illustrated in Figure 5. Trypsin molecules form complexes with Zn^2+^, predominantly through the coordination of amide groups in the trypsin backbone. Meanwhile, precipitations of Zn_3_(PO_4_)_2_ formed in the PBS buffer solution. During the growth process, large agglomerates of trypsin molecules and monolayer nanosheets formed. In the assembly step, trypsin induced nucleation of the Zn_3_(PO_4_)_2_ crystals to form multilayer nanosheets and act as an “adhesive” to bind the petals together. Due to the rapid precipitation reaction and the high content of inorganic components, the generated nanocrystals and complexes assembled and grew on the crystal surface according to the growth mode of inorganic compounds. As the reaction continued, flower-like trypsin/Zn_3_(PO_4_)_2_ hybrid nanoparticles were obtained.

### 2.4. Enzyme Activity of Trypsin/Zn_3_(PO_4_)_2_ Hybrid Nanoflowers

The catalytic activity of trypsin/Zn_3_(PO_4_)_2_ hybrid nanoflowers and free trypsin were evaluated using casein as a substrate. Firstly, the effects of temperature and pH on enzyme activity of the nanoflower and free trypsin were examined, as show in Figure 6a,b. The enzyme activity of trypsin/Zn_3_(PO_4_)_2_ hybrid nanoflowers and free trypsin was investigated in the pH range of 7 to 10 (Figure 6a). The enzyme activity of trypsin/Zn_3_(PO_4_)_2_ was consistently higher than that of the free trypsin at different pH values. In addition, the relative enzyme activity of trypsin/Zn_3_(PO_4_)_2_ was more than 70% in the pH range of 7.5–10. The effect of temperature on enzyme activity of trypsin/Zn_3_(PO_4_)_2_ hybrid nanoflowers and free trypsin was determined in the temperature range of 33 °C to 55 °C (Figure 6b). The enzyme activity of trypsin/Zn_3_(PO_4_)_2_ was also higher than that of the free trypsin with increasing temperature, and optimal trypsin activity was at 45 °C. These results indicate that trypsin/Zn_3_(PO_4_)_2_ hybrid nanoflowers exhibited better enzymatic activity over wider pH and temperature ranges than free trypsin, which was attributed to the limited three-dimensional conformational changes (flip, fold, etc.) of trypsin after being immobilized as nanoflowers [1,34]. Enzyme immobilization by metal ion coordination can effectively improve the catalytic activity of the enzyme, and metal ion coordination limits its conformational changes, making it easier to maintain the structure than in free enzymes. Further, a suitably conceived structure can make the active center more exposed, which may be the main reason for its stable and high activity.

The storage and operational stability of immobilized enzyme is an important feature for its potential application in industry. Thus, the storage stability of trypsin/Zn_3_(PO_4_)_2_ hybrid nanoflowers at 45 °C and 4 °C in buffer solution was performed, as shown in Figure 6c,d. The hybrid nanoflowers showed higher stability than free trypsin at 45 °C for 120 min (Figure 6c). The thermal stability of hybrid nanoflowers gradually became more prominent after 30 min. Further, the storage stability of hybrid nanoflowers was evaluated at 4 °C for a week (Figure 6d). The storage stability of the hybrid nanoflowers exhibited better stability than trypsin, especially after 24 h.

Reusability is an essential indicator to measure the industrial prospect of immobilized enzymes, and it is also the performance to maximize its advantages. As shown in Appendix A, the enzyme activity of trypsin/Zn_3_(PO_4_)_2_ hybrid nanoflowers was not significantly weakened after 12 cycles, and the enzyme still retained 80.1% of its initial activity. The reasons for the decrease of enzyme activity were the micro loss of trypsin in the repeated process, and long-term operation at catalytic temperature.

## 3. Methods and Materials

### 3.1. Materials

Sodium chloride (NaCl, 99.5%), sodium phosphate dibasic dodecahydrate (Na_2_HPO_4_·12H_2_O, 99%), potassium chloride (KCl, 99.5%), potassium phosphate dibasic anhydrous (KH_2_PO_4_, 99%) and zinc acetate dihydrate were purchased from Sinopharm Chemical Reagent Co., Ltd., Xi’an, China. Trichloroacetic acid (TCA), tyrosine, casein, *N,N*-Dimethylformamide (DMF) and tetrahydrofuran (THF), 1,4-dioxane were purchased from Aladdin Reagent Co., Ltd., Shanghai, China. Trypsin was purchased from Tci Reagent Co., Ltd., Shanghai, China. Deionized (DI) water was used for all aqueous solutions and tests.

### 3.2. Preparation of the PBS Buffer

NaCl (8.00 g), Na_2_HPO_4_ (1.44 g), KCl (0.20 g) and KH_2_PO_4_ (0.24 g) were dissolved in 1 L deionized water and cooled to room temperature. The concentration of the buffer solution was 0.01 M.

### 3.3. Synthesis of Trypsin/Zn_3_(PO_4_)_2_ Hybrid Nanoflowers

Trypsin (0.05 g) was dissolved in 20 mL of PBS Buffer and stirred at 30 °C. Under mechanical stirring, 24 mL (2.5 wt%) zinc acetate solution was added into the solution and continually stirred for 2 h. After that, a white suspension was obtained and separated by centrifugation. Then, the white solid was washed three times with water. Finally, the trypsin/Zn_3_(PO_4_)_2_ hybrid nanoflowers were obtained by removing water with a freeze dryer.

The reaction temperatures were 30 °C, 40 °C and 50 °C, respectively. The amounts of trypsin and Zn(OAc)_2_·2H_2_O are shown in Appendix A, and the parallel experiments were all single-variable.

### 3.4. Activity Assays of Free Trypsin and Trypsin/Zn_3_(PO_4_)_2_

Enzyme activity of free trypsin was determined with casein as substrate. The measurement was developed by Esmaeil with some minor modifications. The typical process was as follows: 1% (*w*/*v*) casein and trypsin (50 μg/mL) were dissolved in 500 μL PBS buffer (pH = 7.5). After incubation at 25 °C for 10 min, the reaction was terminated with 500 μL 10% trichloroacetic acid (TCA), and the mixture was separated by centrifugation at 20,000 rpm for 10 min. The absorbance of the supernatant was measured at 280 nm. One unit of enzymatic activity was defined as the amount of enzyme hydrolyzing 1 μmol tyrosine per minute at 25 °C. The enzyme activity of free trypsin was calculated by the concentration of tyrosine.

In the case of trypsin/Zn_3_(PO_4_)_2_, the activity measurements and conditions were similar to those of free trypsin, and trypsin/Zn_3_(PO_4_)_2_ was prepared by reacting 0.05 g trypsin and 0.56 g zinc acetate at 30 °C for 120 min in 20 mL PBS buffer solution.

The thermostability of temperature on free trypsin and trypsin/Zn_3_(PO_4_)_2_ activity was determined in the temperature range 33–55 °C and for different incubation times (0–60 min) at 45 °C. The effect of pH on activity of free and immobilized trypsin was assayed in different PBS buffers of pH ranging from 7.0 to 10.0. These tests were performed in triplicate. The durability of free trypsin and trypsin/Zn_3_(PO_4_)_2_ was investigated at 4 °C for a week. The reusability of trypsin/Zn_3_(PO_4_)_2_ hybrid nanoflowers was conducted at 37 °C and 12 repeated cycles.

### 3.5. Characterization

The morphologies of trypsin/Zn_3_(PO_4_)_2_ hybrid nanoflowers were observed by scanning electron microscopy (SEM, FEI Verios G4, Waltham, Thermo Fisher Scientific Ltd, Waltham MA, USA) and transmission electron microscope (TEM, JEM-ARM300F, JEOL Ltd, TOKYO, JPN). Fourier transform infrared (FT-IR) spectra of the nanoflowers was conducted with an FT-IR (FTIR, TENSOR27, Bruker Ltd, Billerica, Massachusetts, USA). The crystal structures of the nanoflowers were determined by X-ray powder diffraction (XRD, Thermo Scientific 7000, Thermo Fisher Scientific Ltd, Waltham, MA, USA). Thermogravimetric analysis (TGA, Mettler Toledo Q50) was carried out in an O_2_ atmosphere. The temperature range was 50–500 °C, and the heating rate was 10 °C/min. Specific surface areas and pore size distribution were performed by Brunauer–Emmet–Teller (BET) using an N_2_ physisorption analyzer (Tristar3020, Micromeritics Ltd, Norcross, Georgia, USA). The concentration of tyrosine was calculated by absorbance, which was obtained by a UV–vis spectrophotometer with a dual optical path (Shanghai Unico Ltd, Shanghai, China).

## 4. Conclusions

In summary, trypsin-incorporated Zn_3_(PO_4_)_2_ hybrid nanoflowers were successfully prepared under mild conditions using Zn^2+^ as the intermediate medium, which precipitated with phosphate and coordinated with trypsin. The concentration and ratio of raw materials and reaction temperature are the core factors influencing the morphology of hybrid nanoflowers. The formation mechanism of trypsin/Zn_3_(PO_4_)_2_ hybrid nanoflowers was clarified by the intuitive observation results of morphology tracking. Trypsin/Zn_3_(PO_4_)_2_ hybrid nanoflowers as an immobilized enzyme demonstrated excellent enzyme activity, including exceptional stability to pH and temperature, storage stability and reusability, compared with free trypsin. The optimal catalytic pH and temperature are 8.5 and 45 °C, respectively. We are convinced that our proposed strategy for fabricating trypsin/Zn_3_(PO_4_)_2_ hybrid nanoflowers will contribute to the extensive application of immobilized enzyme in industrial biocatalysis.

## Figures and Tables

**Figure 1 ijms-23-11853-f001:**
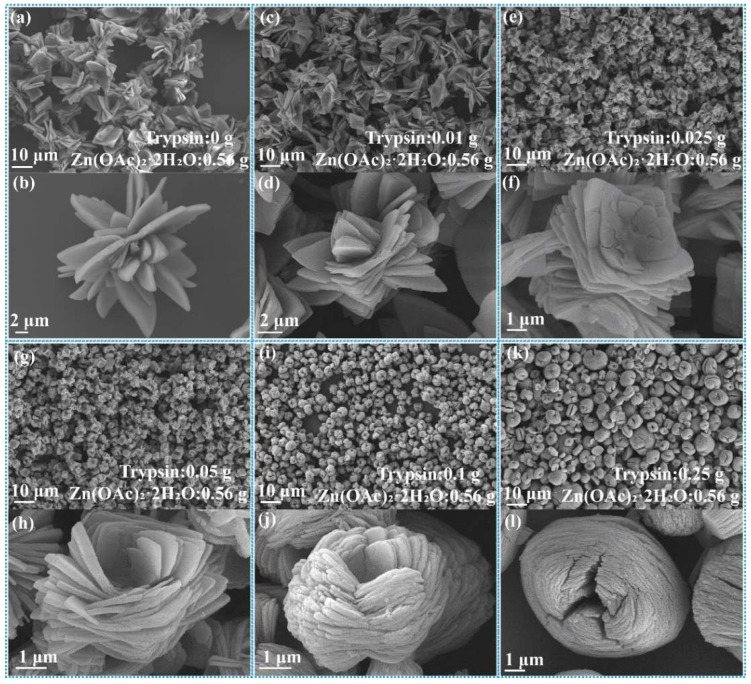
SEM images of trypsin/Zn_3_(PO_4_)_2_ hybrid nanoflowers prepared by adding the following amounts of trypsin: (**a**,**b**) 0 g, (**c**,**d**) 0.01 g, (**e**,**f**) 0.025 g, (**g**,**h**) 0.05 g, (**i**,**j**) 0.1 g and (**k**,**l**) 0.25 g. Zn(OAc)_2_·2H_2_O was 0.56 g.

**Figure 2 ijms-23-11853-f002:**
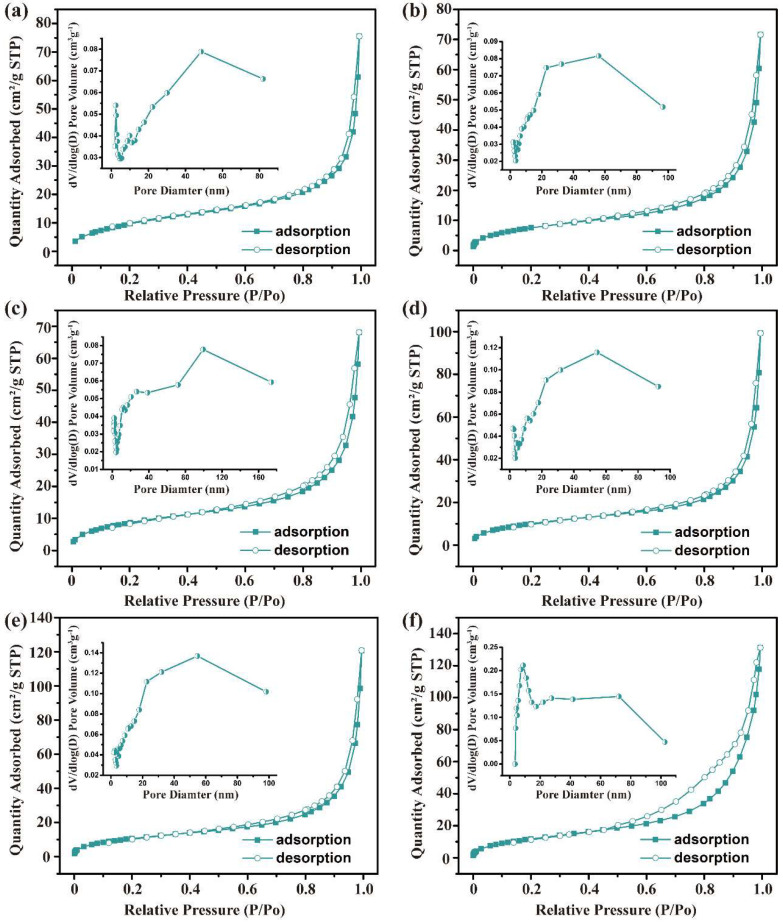
N_2_ absorption–desorption isotherm of trypsin/Zn_3_(PO_4_)_2_ hybrid nanoflowers prepared by adding the following amounts of trypsin: (**a**) 0 g, (**b**) 0.01 g, (**c**) 0.025 g, (**d**) 0.05 g, (**e**) 0.1 g and (**f**) 0.25 g. The inset is pore-size distribution curve.

**Figure 3 ijms-23-11853-f003:**
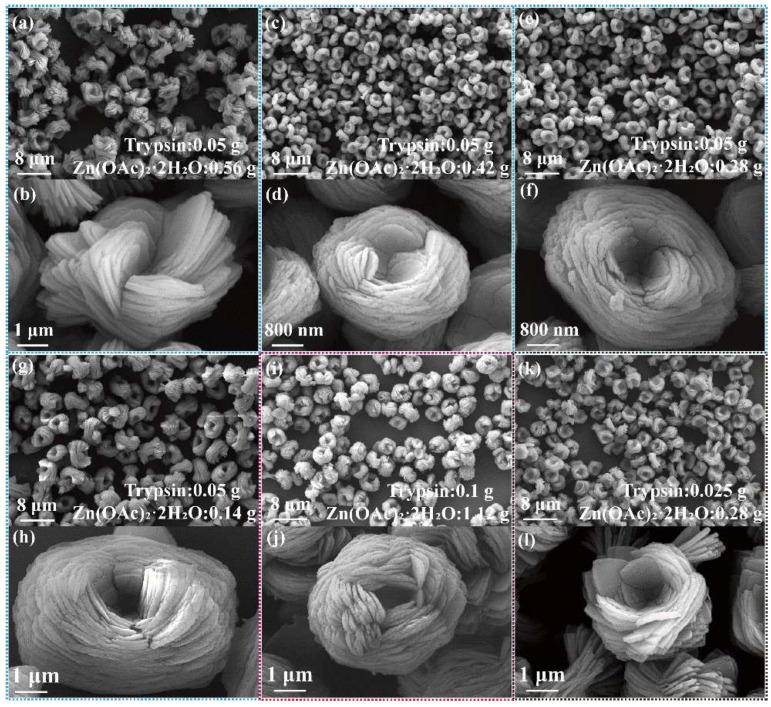
Effect of amount of Zn^2+^ added on trypsin/Zn_3_(PO_4_)_2_ hybrid nanoflowers; the dosage of Zn(OAc)_2_·2H_2_O was (**a**,**b**) 0.56 g, (**c**,**d**) 0.42 g, (**e**,**f**) 0.28 g and (**g**,**h**) 0.14 g. Trypsin amount was 0.05 g. The concentration of raw material is doubled (**i**,**j**) and halved (**k**,**l**).

**Figure 4 ijms-23-11853-f004:**
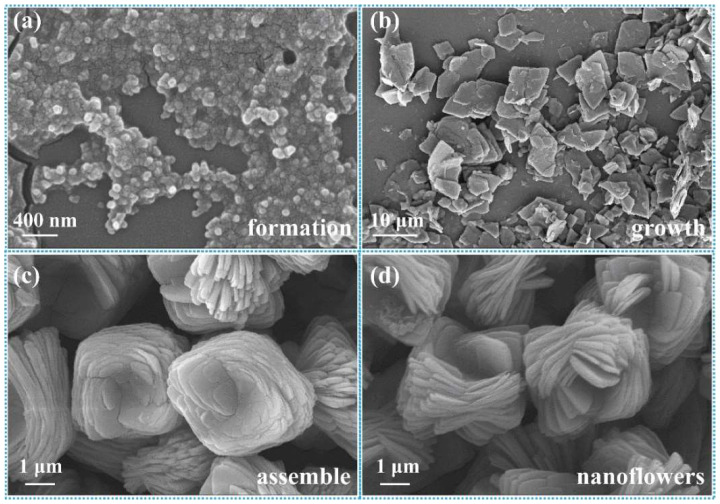
SEM images of trypsin/Zn_3_(PO_4_)_2_ hybrid nanoflowers obtained under different reaction times: (**a**) 10 min, (**b**) 20 min, (**c**) 60 min and (**d**) 120 min.

**Figure 5 ijms-23-11853-f005:**
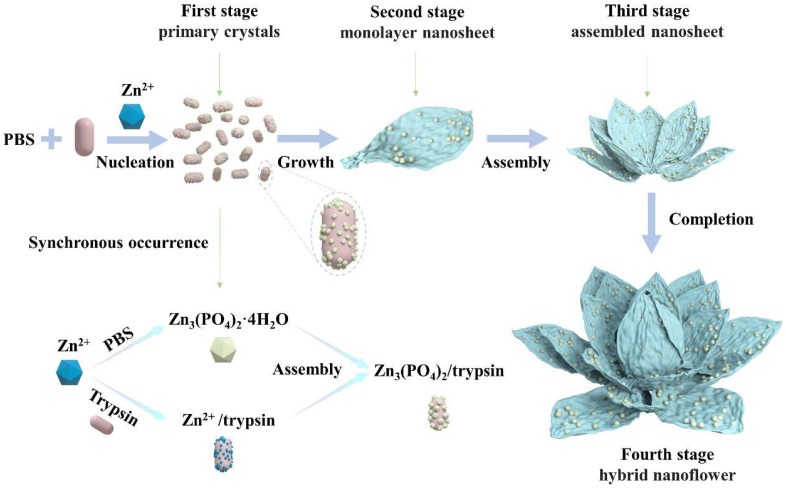
Formation of trypsin/Zn_3_(PO_4_)_2_ hybrid nanoflowers.

**Figure 6 ijms-23-11853-f006:**
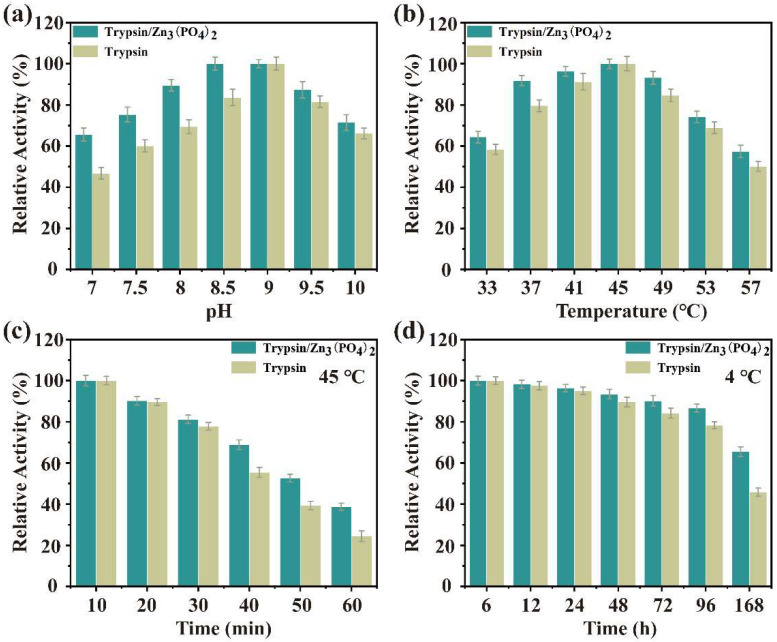
Effect of pH (**a**) and temperature (**b**) on the catalytic activity of trypsin/Zn_3_(PO_4_)_2_ hybrid nanoflowers; thermo (**c**) and storage (**d**) stability of trypsin/Zn_3_(PO_4_)_2_ hybrid nanoflowers. Data are presented as mean ± SD (n = 3).

## Data Availability

Not applicable.

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
