# Peer review of "Trypsin/Zn3(PO4)2 Hybrid Nanoflowers: Controlled Synthesis and Excellent Performance as an Immobilized Enzyme"

_ijms, 2022, doi:10.3390/ijms231911853_

Round 1
Reviewer 1 Report
The authors’ aim was trypsin immobilization in a new nanoflower system.
The literature presents the necessary research background that formed the basis of the experiments carried out. The experiments were logically structured, and the immobilized biocatalysts were analyzed with numerous analytical tests. Thus, the publication presents useful results from this field of enzyme immobilization.
Some comments and additional test suggestions you can see below:
1. In line 63 “And” do not necessary to the beginning of the sentence.
2. In line 65 please omit the “the” in the “by tracking the its…” sentence.
3. In chapter 2.2 please add also the concentrations to the buffer solution description.
4. The Ac group is the CH3-C(=O)- group, not the CH3-COO, so the Zn(Ac)2 is not correct, please use Zn(OAc)2 in the text and the figures.
5. The reviewer thinks besides the morphological properties the activity of the immobilized biocatalysts is also very important. Were the biocatalytic activity measured during the optimization steps (trypsin amount and Zn-salt amount optimization)? Biocatalytic activities and activity yield would be important additional data and maybe some correlation will be detected between the activities and pore size.
6. Please defined which method was used for the preparation of the immobilized enzyme used in section 3.4.
7. In figure 6. there are error bars on the diagrams. How many measurements were taken? Please add this information to the figure capture.
8. In figure capture at Figure 6. there are A and B marks, but on the picture, there are no A and B. Also, there are four diagrams but there are only two letter designations.
Reviewer 2 Report
The paper by Wang et al. present preparation of trypsin/Zn3(PO4)2 nanoflowers. The object is research is clearly presented and well characterised.
However, I would like authors to discuss in more details the increase the activity of trypsin in the nanoflowers. The authors propose that limited three conformational changes are the factor stimulating the activity. Could they elaborate this on the basis on know trypsin structure and molecular mechanism? Could they say something about Km/ kcat of the free and immobilised enzyme? Additionally, I would imagine, e.g. that substrate accessibility will be lower for immoblised enzymes. Why this is not a factor?
There are some small language problems (typos, etc) which also should be corrected.
Round 2
Reviewer 1 Report
Thanks to the authors for correcting the manuscript well and thoroughly documenting it.
A publication presents good and useful results was published.